# 1,25(OH)_2_D3 Differently Modulates the Secretory Activity of IFN-DC and IL4-DC: A Study in Cells from Healthy Donors and MS Patients

**DOI:** 10.3390/ijms24076717

**Published:** 2023-04-04

**Authors:** Isabella Sanseverino, Arturo Ottavio Rinaldi, Cristina Purificato, Antonio Cortese, Enrico Millefiorini, Maria Cristina Gauzzi

**Affiliations:** 1National Center for Global Health, Istituto Superiore di Sanità, 00161 Rome, Italy; 2Multiple Sclerosis Center, Sapienza University of Rome, 00161 Rome, Italy

**Keywords:** type I IFN, vitamin D, chemokine, cytokine, monocyte-derived dendritic cell, multiple sclerosis, immune pathophysiology, modifiable environmental factor

## Abstract

Immune mechanisms play an essential role in driving multiple sclerosis (MS) and altered trafficking and/or activation of dendritic cells (DC) were observed in the central nervous system and cerebrospinal fluid of MS patients. Interferon β (IFNβ) has been used as a first-line therapy in MS for almost three decades and vitamin D deficiency is a recognized environmental risk factor for MS. Both IFNβ and vitamin D modulate DC functions. Here, we studied the response to 1,25-dihydoxyvitamin D3 (1,25(OH)_2_D3) of DC obtained with IFNβ/GM-CSF (IFN-DC) compared to classically derived IL4-DC, in three donor groups: MS patients free of therapy, MS patients undergoing IFNβ therapy, and healthy donors. Except for a decreased CCL2 secretion by IL4-DC from the MS group, no major defects were observed in the 1,25(OH)_2_D3 response of either IFN-DC or IL4-DC from MS donors compared to healthy donors. However, the two cell models strongly differed for vitamin D receptor level of expression as well as for basal and 1,25(OH)_2_D3-induced cytokine/chemokine secretion. 1,25(OH)_2_D3 up-modulated IL6, its soluble receptor sIL6R, and CCL5 in IL4-DC, and down-modulated IL10 in IFN-DC. IFN-DC, but not IL4-DC, constitutively secreted high levels of IL8 and of matrix-metalloproteinase-9, both down-modulated by 1,25(OH)_2_D3. DC may contribute to MS pathogenesis, but also provide an avenue for therapeutic intervention. 1,25(OH)_2_D3-induced tolerogenic DC are in clinical trial for MS. We show that the protocol of in vitro DC differentiation qualitatively and quantitatively affects secretion of cytokines and chemokines deeply involved in MS pathogenesis.

## 1. Introduction

Multiple sclerosis (MS) is an immune-mediated disorder of the central nervous system (CNS) characterized by inflammation and demyelination as well as axonal and neuronal degeneration [1].

Vitamin D deficiency is currently considered one of the main environmental MS risk factors. This hypothesis, first proposed over 30 years ago to explain the geographical distribution of disease prevalence, gained credibility after the discovery of vitamin D immunomodulatory effects, and it is now supported by consistent epidemiological, clinical, and genetic data [2,3,4]. The number of observations supporting a protective effect of vitamin D in MS fostered the idea that vitamin D deficiency might be an important modifiable risk factor, and several clinical trials have been performed to evaluate the efficacy of vitamin D in relapsing-remitting MS (RRMS), also in combination with interferon β (IFNβ) [5,6,7]. In spite of the solid evidence establishing a causal link between vitamin D deficiency and the risk of MS, disease activity, and progression, clinical trials have largely been inconclusive [5,8]. There are still uncertainties and many unanswered questions, including how vitamin D exerts a protective effect, and which are the relevant cell targets.

Dendritic cells (DC) are professional antigen presenting cells that play key roles in inducing immune responses and immune tolerance. Evidence indicates that DC contribute to the pathogenesis of MS. DC are found within MS lesions and are functionally abnormal in patients with MS. In particular, both myeloid and plasmacytoid DC (pDC) are enriched in the cerebrospinal fluid (CSF) of MS patients, especially at the time of acute relapse [9], and immature and mature myeloid DC are present in MS lesions in the CNS. Importantly, these DC engulf myelin components and interact with proliferating T cells [10,11]. Given their perivascular location, it was proposed that intracerebral DC derive from DC and/or DC precursors recruited from the blood, which differentiate/mature in the inflamed CNS environment [12].

DC might also provide an avenue for therapeutic intervention. Cell-based therapies are of increasing interest in the treatment of MS [13], and because of the key role of DC in the induction and maintenance of central and peripheral tolerance, in vitro induced tolerogenic DC (tolDC) show some promise [14]. Different strategies can be used to generate tolDC, including pharmacologic intervention with 1,25-dihydroxyvitamin D3 (1,25(OH)_2_D3), the biologically active vitamin D metabolite [15]. Indeed, clinical trials with autologous tolDC generated in vitro with 1,25(OH)_2_D3 have been performed [16].

Blood monocytes can be in vitro differentiated into DC with different cytokine cocktails. The combination GM-CSF/IL4 represents the first described and, by far, the most extensively characterized [17]. However, a three-day culture of monocytes with type I IFN, in combination with GM-CSF, can also drive monocyte differentiation into fully functional, partially mature DC (IFN-DC) [18]. IFN-DC exhibit distinct phenotypic and functional features with respect to DC obtained with IL4 and GM-CSF (IL4-DC). IFN-DC express higher levels of costimulatory molecules and variable amounts of the maturation marker CD83 in the absence of further maturation stimuli [19] and share phenotypic and functional hallmarks, such as the expression of CD123–with pDC, the blood DC subset specialized in type I IFN production in response to virus infection and other danger signals [20]. Compared to IL4-DC, IFN-DC express higher levels of CCR5 and exhibit an enhanced response to the CCR5 ligands CCL5, CCL3, CCL4. A considerable fraction of IFN-DC also expresses CCR7 and has improved migratory response to CCL19 [21]. IFN-DC efficiently initiate adaptive immune responses, in vitro as well as in animal models, promoting a Th1 biased CD4 response, the expansion of CD8 effector T cells as well as IgG1 isotype antibodies production. For these characteristics, IFN-DC are promising candidates for the development of DC-based immunotherapies in cancer [22].

The IFN-driven pathway of DC generation may also be of relevance in vivo, since type I IFN is locally produced in physiological condition and production is highly enhanced in pathologies, such as viral infections or inflammatory diseases. In particular, such a DC differentiation pathway might be relevant in MS, either as a pathogenic mechanism and/or as a protective pathway during IFN therapy, which still represent one of the first-line MS treatments [23]. Notably, we previously observed that in the IFN-DC model, the suppression of differentiation/activation by 1,25(OH)_2_D3 is associated with a potent impairment of their chemotactic response [24], which could be potentially protective with respect to MS lesions.

In this study, we compared the expression levels of vitamin D receptor (*VDR*) and of its responsive gene *CYP24A1* (i.e., the gene encoding the catabolic enzyme inactivating 1,25(OH)_2_D3) as well as the secretory activity of IFN-DC and IL4-DC obtained from healthy donors and MS patients, either free of therapy or undergoing IFNβ therapy.

## 2. Results

### 2.1. IFN-DC Express Higher Levels of VDR mRNA Than IL4-DC

The biological effects of 1,25(OH)_2_D3 are mediated by VDR, a member of the superfamily of nuclear hormone receptors. The enzyme CYP24A1, which catalyzes the inactivation of 1,25(OH)_2_D3, is encoded by a primary vitamin D-responsive gene [25]. Therefore, we sought to characterize the expression of these two important vitamin D effector molecules in our cell models (depicted in Figure 1).

As shown in Figure 2, both IFN-DC (Figure 2a) and IL4-DC (Figure 2b) from the three donor groups constitutively express *VDR* mRNA. The presence of 1,25(OH)_2_D3 did not significantly change *VDR* mRNA expression levels in IFN-DC in any donor group, accordingly to what was previously observed in IFN-DC obtained from buffy coats of healthy donors [24]. A trend toward a reduction—although not statistically significant—was observed in IL4-DC from the three groups of donors in response to 1,25(OH)_2_D3. No significant differences were observed among the three donor groups, although a trend toward higher basal expression in IFN-DC obtained from MS-I patients is observable. Interestingly, after pooling data obtained from the three donor groups, we did observe a significantly higher *VDR* expression in IFN-DC compared to IL4-DC (Figure 2c).

The basal expression of *CYP24A1* mRNA was generally under the detection limit but it was readily induced in the presence of 1,25(OH)_2_D3 (Figure 2d,e) at comparable levels in the three donor groups and in the two cell models (Figure 2f).

### 2.2. 1,25(OH)_2_D3 Differently Modulates the Secretion of Pro- and Anti-Inflammatory Mediators in IFN-DC and IL4-DC

Soluble mediators of inflammation released by activated immune cells are key regulators of the autoimmune response in MS. Proinflammatory cytokines and chemokines such as IL1β, IL6, IL8, and TNFα promote immune cell infiltration through the blood–brain barrier (BBB), leading to demyelination and axonal damage [26], while anti-inflammatory molecules such as IL10 could reduce the immune damage of the CNS [27]. We thus analyzed the basal and 1,25(OH)_2_D3-induced secretion of these soluble mediators in our cell models.

IFN-DC from the three groups of donors constitutively produced the pleiotropic cytokine IL6, and no significant changes were observed when they were differentiated in the presence of 1,25(OH)_2_D3 (Figure 3a). Conversely, IL6 levels were generally under the detection limit in supernatants of untreated IL4-DC, but high levels of IL6 secretion were induced in the presence of 1,25(OH)_2_D3 in IL4-DC from all three donor groups (Figure 3b). Both IFN-DC and IL4-DC also produced low but detectable levels of the soluble receptor sIL6R (Figure 3c,d). Interestingly, 1,25(OH)_2_D3 treatment induced sIL6R in IL4 DC (Figure 3d), but not in IFN-DC (Figure 3c). The induction was significant in IL4-DC from the H group, but a trend toward increased secretion was also observed in IL4-DC from MS and MS-I patients.

IFN-DC produced low but detectable levels of TNFα, which diminished in the presence of 1,25(OH)_2_D3, in cells from H and MS donors (reaching in the latter group statistical significance) (Figure 3e). A trend toward TNFα induction by 1,25(OH)_2_D3 was observed in IFN-DC from MS-I patients, although the difference between untreated and 1,25(OH)_2_D3-treated cells was not statistically significant (Figure 3e). IL4-DC constitutively produced comparable low levels of TNFα as IFN-DC, but no effect of 1,25(OH)_2_D3 treatment was observed in any donor group (Figure 3f).

IFN-DC constitutively produced a high level of the inflammatory chemokine IL8 which was strongly reduced in the presence of 1,25(OH)_2_D3 (Figure 3g). Low secretion of IL8 in the presence or in the absence of 1,25(OH)_2_D3 was observed in the three groups of IL4-DC (Figure 3h).

IL1β was secreted at very low levels, irrespectively of the cell model/group of donors and was not modulated by 1,25(OH)_2_D3 (Appendix A).

No significant donor group effect was found in constitutive or 1,25(OH)_2_D3-induced secretion of IL6, sIL6R, TNFα and IL8 by testing IFN-DC or IL4-DC from the three groups of donors with a two-way ANOVA (Appendix A).

We next analyzed the anti-inflammatory cytokine IL10. Both IFN-DC and IL4-DC from the three groups of donors constitutively produced low but detectable levels of IL10 (Figure 4). However, the two cell models differed in their response to 1,25(OH)_2_D3: the hormone significantly reduced IL10 in culture supernatants of IFN-DC (all three donor groups) (Figure 4a), while no significant differences were observed in culture supernatants of IL4-DC (Figure 4b).

Although no significant overall group effect was found by two-way ANOVA test (Appendix A), multiple comparisons in post hoc analysis revealed a significant difference in the basal IL10 secretion of MS-IFN-DC compared to H-IFN-DC (Table 1).

### 2.3. 1,25(OH)_2_D3 Differently Modulates Secretion of CCR2 and CCR5 Ligands in IFN-DC and IL4-DC

We next analyzed the secretion of CCL2 and CCL5, which play complex roles in MS being involved in both CNS damage and repair [28]. IFN-DC constitutively produced high levels of CCL2, further enhanced when the cells were differentiated in the presence of 1,25(OH)_2_D3 (Figure 5a). The induction is statistically significant for IFN-DC obtained from MS patients, either in therapy or not. As we previously described [29], 1,25(OH)_2_D3 strongly induced CCL2 secretion in IL4-DC, which constitutively produced low levels of the chemokine (Figure 5b).

We confirmed here that IL4-DC from the MS group secrete less CCL2 in response to 1,25(OH)_2_D3, as revealed by multiple comparison post hoc analysis of ANOVA test (Table 2 and Appendix A).

### 2.4. IFN-DC Secrete High Levels of MMP9, Which Are Strongly Reduced by 1,25(OH)_2_D3

Finally, we analyzed the secretion of matrix-metalloproteinase 9 (MMP9), an enzyme contributing to neuroinflammation, found altered in the serum and CSF of RRMS patients [30,31]. IFN-DC constitutively secreted high levels of MMP9, which were strongly reduced in the presence of 1,25(OH)_2_D3 in all experimental groups (Figure 6a). Lower constitutive levels of MMP9 were observed in IL4-DC from the three groups, which were further reduced in the presence of 1,25(OH)_2_D3 in the MS group (Figure 6b).

## 3. Discussion

IFNβ treatment is a first line therapy for RRMS [32], and vitamin D deficiency is an established risk factor for MS disease and severity [4]. Both molecules act on DC, in vitro and in vivo [22,33,34,35], modulating their functional activity. However, the combined action of vitamin D and IFNβ in these cells and its potential relevance in MS have been poorly investigated. Here, we studied the response to 1,25(OH)_2_D3 in IFN-DC and IL4 DC, obtained from RRMS patients undergoing IFN therapy or not, as well as healthy donors.

We did not find any significant difference among DC from the three groups of subjects in the expression of two key effectors of the vitamin D response, VDR and CYP24A1, suggesting that DC from MS patients and H controls are similarly equipped to respond to 1,25(OH)_2_D3. However, the higher VDR level observed in IFN-DC, together with the trend toward higher expression in MS-I-IFN-DC, suggest a possible synergy between vitamin D and IFNβ. Interestingly, clinical trials of vitamin D in combination with IFNβ have been performed (ClinicalTrials.gov, Identifier NCT01198132; NCT01339676; NCT01285401) and at least one is ongoing (ClinicalTrials.gov, Identifier: NCT02903537). Data from completed trials are not in contrast with such a synergy. Although the primary endpoints were not generally met, secondary endpoints [6,7,36] were in favor of the vitamin D3-treated group, leaving open and worthwhile of further investigation the possibility of a protective effect of vitamin D as add-on therapy to IFNβ.

The analysis of the secretion of soluble mediators in response to 1,25(OH)_2_D3 did not reveal major defects in DC from MS donors compared to those from healthy donors, whatever the cell model, with the exception of a decreased secretion of CCL2 by IL4-DC from the MS group (confirming our previous observations [29]) and an increased basal IL10 secretion in IFN-DC from the MS group. The differences were, however, only quantitative and modest in size.

On the contrary, the comparison of the basal and 1,25(OH)_2_D3-induced cytokine/chemokine secretion in IFN-DC versus IL4-DC revealed qualitative and quantitative differences in the production of several soluble mediators involved in MS pathogenesis.

A major difference in the response to 1,25(OH)_2_D3 between the two DC models was the secretion of IL6, which was potently induced only in IL4-DC. The observation is in line with previous reports [37,38] in spite of the fact that 1,25(OH)_2_D3 is generally described as an inhibitor of IL6 secretion [39,40,41], likely because it effectively reduces IL6 and other inflammatory cytokines upon activation by Toll-like receptor ligation or other stimuli.

IL6 exerts its cellular effects through two distinct pathways. In the classic pathway, IL6 stimulates target cells via a membrane bound receptor (IL6R), which upon ligand binding associates with the signaling glycoprotein gp130. An additional pathway, known as IL6 trans-signaling, relies on the soluble form of the IL6R (sIL6R) which allows IL6R-negative cells (expressing the common subunit gp130) to respond to IL6 [42]. The induction of both IL6, and to a lesser extent, sIL6R receptor in IL4-DC suggests that both pathways could be activated in an autocrine/paracrine way by 1,25(OH)_2_D3. IL6 is considered an important pro-inflammatory cytokine contributing to MS pathology [26,43,44] based on preclinical studies in animal models [45,46] and on clinical observations of an association between higher CSF levels of IL6 and severity of disease [43,44]. IL6 is also known (together with IL1β) to enhance T helper cell differentiation toward the TH17 subset, which is pathogenic in MS, but IL6 may also have homeostatic roles and it is endowed with regenerative or anti-inflammatory activities [46,47]. Recent data point to the role of vitamin D induced-IL6 in establishing tolerogenesis/dampening inflammatory responses via the activation of a IL6-JAK-STAT3 pathway, which in turn promotes production of IL10 [48,49]. Current data are not sufficient to establish whether IL6 induction by 1,25(OH)_2_D3 in IL4-DC could be of relevance in MS pathogenesis or DC-based immunotherapy.

TNFα is another inflammatory cytokine involved in pathological hallmarks of MS, including immune dysregulation, demyelination, synaptopathy and neuroinflammation [50]. In the presence of 1,25(OH)_2_D3, TNFα secretion by IFN-DC from H and MS, but not MS-I donors diminished. This suggests a modest—not statistically significant—effect of IFNβ therapy on the in vitro secretion of this cytokine.

IFN-DC and IL4-DC also strongly differed in basal secretion of IL8, another pro-inflammatory mediator. IL8 is a chemo-attractant for neutrophils and monocytes which triggers their firm adhesion to the endothelium and may contribute to MS pathogenesis via promoting transmigration of lymphocytes across the BBB. Serum IL8 and IL8 secretion from PBMCs are augmented in MS patients [51]. Although the high basal IL8 levels secreted by IFN-DC are potentially detrimental in MS, they were strongly reduced in the presence of 1,25(OH)_2_D3.

A further difference between the two cell models was the secretion of IL10, an anti-inflammatory cytokine found deregulated in MS. IL10 secretion by PBMC is decreased prior to relapse and increased during remission, suggesting it may play an important role in recovery [26,52]. In line with the trend toward IL10 induction that we observed in IL4-DC, 1,25(OH)_2_D3 was previously reported to enhance IL10 secretion by IL4-DC, especially upon activation by stimuli such as CD40 ligation, cytokine cocktails or Toll-like receptor ligands [38,53]. Conversely, the hormone significantly reduced IL10 in culture supernatants of IFN-DC from all three donor groups. Furthermore, we observed a slightly—but statistically significant—higher basal IL10 secretion in MS-IFN-DC compared to H-IFN-DC. Whether this could be associated to MS subjects being in the recovery phase after the last relapse could not be excluded.

CCR5 and CCR2 ligands mediate inflammation, leukocyte recruitment to active CNS lesions, and axonal damage, but may also be involved in tissue repair [28]. We confirm here that 1,25(OH)_2_D3 induces high levels of CCL2 in IL4-DC and extend the observation to the IFN-DC model. Moreover, in the IL4-DC model, it also induces CCL5, although to a lower level and without reaching statistical significance. These findings could appear incoherent with the anti-inflammatory activity of 1,25(OH)_2_D3 as a pathogenic role of CCL2/CCR2, as well as of CCR5 and its ligands, which in MS is widely recognized [54,55]. However, the involvement of these chemokine/receptors axes in MS is multifaceted [28]. CNS invading Th1 cells and monocytes, as well as CNS resident innate immune cells such as microglia and astrocytes, co-express CCR2 and CCR5, and the ligands CCL2, CCL3, CCL4 and CCL5 are up-regulated in MS lesions. These chemokines may recruit leukocyte to active lesions and amplify local autoimmune/inflammatory responses ultimately leading to demyelination and axonal damage. CCL2 and CCR5 ligands also contribute to remyelination and damage repair. CCR5^+^CCR2^+^ macrophages are essential for the clearance of myelin debris. CCL2 also enhances oligodendrocyte precursor mobility, enabling them to populate demyelinated lesions, where they differentiate into myelin sheath-forming oligodendrocytes ([28] and references therein). Whether the reduced CCL2 secretion in MS patients has relevance in MS clinical course and/or response to vitamin D supplementation remains to be established.

MMPs are proteolytic enzymes contributing to tissue remodeling and development and have been involved—MMP9 in particular—in the BBB destroy those which occurs in MS [56,57]. Increased CSF and serum levels of MMP9 were observed in MS patients [56,57] and MMP9 level has been proposed as a biomarker of MS clinical course [30]. A recent meta-analysis [57] revealed the significant association of a MMP9 single nucleotide polymorphism with increased disease risk. Additionally, MMPs may control extracellular pools of inflammatory cytokines and chemokines [31]. MMP9 mediated proteolysis has been reported to allow the release of surface bound TNFα forms [58] and selective cleavage of IL8 or CCL2 resulting in modulation of their biological activity [59]. We observed a trend toward a higher basal MMP9 secretion by IL4-DC from MS patients (either in therapy or not) although the difference with H-IL4-DC did not reach statistical significance. The presence of 1,25(OH)_2_D3, however, reduced this secretion to the same levels observed in healthy donors. A down-modulating activity of 1,25(OH)_2_D3 on MMPs secretion by PBMC was already reported in the context of *Mycobacterium tuberculosis* infection [60] or of skin inflammation following UV exposure [61].

DC may contribute to the pathogenesis of MS, but also provide an avenue for therapeutic intervention. 1,25(OH)_2_D3-induced tolDC are emerging as therapeutic tools and seem to be a promising strategy for restoring tolerance in MS [14,62]. Phase I clinical trials with tolDCs in MS patients have shown these therapies are safe and well tolerated, with no relevant adverse effects [63]. Two active clinical trials with tolDC (IL4-DC) generated with 1,25(OH)_2_D3 are registered at ClinicalTrials.gov [MS-tolDC, ClinicalTrials.gov Identifier: NCT02618902; TOLERVIT-MS, ClinicalTrials.gov Identifier: NCT02903537]. Our results showing that IL4-DC from RR-MS patients respond to 1,25(OH)_2_D3 similarly to those of healthy subjects are in agreement with published data [64,65,66] which provided the rationale of using autologous DCs in MS treatment [16,67] and may contribute to reinforce this rationale. We additionally report that IFN-DC from RR-MS and healthy donors showed a similar sensitivity to the in vitro activity of 1,25(OH)_2_D3. This observation may be of relevance in MS as we hypothesized that IFN-DC—with the intrinsic limits of any in vitro system—could mimic an IFN-driven pathway of DC generation occurring in vivo, when type I IFN is locally and abundantly produced, such as in viral infections or inflammatory diseases. Indeed, both these conditions contribute to MS. The Epstein-Barr virus infection emerged as a key environmental factor linked to MS etiopathogenesis [68,69], and inflammation in the CNS is a hallmark of the disease [70].

The IFN-driven DC differentiation pathway might also occur following IFN therapy and be protective. Support to this hypothesis came from the murine model [71] where myeloid cells were shown to mediate the protective effect of IFN on experimental autoimmune encephalomyelitis (EAE). Furthermore, targeting IFN activity selectively to DC through the use of Activity-on-Target IFNs, which are IFN-based immunocytokines coupled to ligands for cell-specific surface markers, induces a robust in vivo tolerization, efficiently protecting against EAE (i.e., DC are among the cells mediating the protective effect of IFN) [72].

Notably, DC differentiated in the presence of type I IFN are partially active and already endowed with immunostimulatory activity, in vitro as well as in animal models, without the need of further activation/maturation stimuli [22]. 1,25(OH)_2_D3 strongly inhibited the expression of potentially detrimental mediators. Indeed, with the exception of CCL2, all cytokines/chemokines analyzed were either unaffected or down-modulated by 1,25(OH)_2_D3 in IFN-DC. Thus, 1,25(OH)_2_D3 may exert a mitigating action on the intrinsic pro-inflammatory characteristics of IFN-DC which are potentially harmful in the context of MS, such as the expression of high levels of IL8 or MMP9.

DC might be among the target cells mediating the clinical benefits of vitamin D or clinical damage due to vitamin D deficiency. Our in vitro analysis of two DC models confirmed cell-type specificity of vitamin D actions [73] and points out that the protocol of in vitro differentiation qualitatively and quantitatively affects the basal and 1,25(OH)_2_D3-induced secretion of cytokines and chemokines deeply involved in MS pathogenesis.

## 4. Materials and Methods

### 4.1. Blood Samples

Peripheral blood (5–7 mL) was collected in EDTA from three groups of donors:MS: RRMS patients free of disease modifying therapies;MS-I: RRMS patients undergoing high dose treatment with IFNβ-1a and/or 1b;H: healthy volunteers with a comparable gender and age distribution.

Clinical and demographic characteristics are summarized in Appendix A (Appendix A). At blood withdrawal, all patients were free of corticosteroid therapy and in the stable phase of the disease, as based on clinical features. The diagnosis of RRMS was established by clinical, laboratory, and MRI parameters, and matched published criteria [74]. All blood donors gave their written informed consent to this study, approved by the Research Ethics Committee of the Istituto Superiore di Sanità (CE-ISS 10/294).

### 4.2. Monocyte Separation and DC Culture

Peripheral blood mononuclear cells (PBMC) were isolated by centrifugation on density gradient (Lymphprep, Stemcell Technology, Vancouver, BC, Canada), and CD14^+^ monocytes purified by immunomagnetic positive selection (MACS monocyte isolation kit, Miltenyi Biotec, Bergisch Gladbach, Germany). To obtain IL4-DC, monocytes were seeded at 1 × 10^6^ cells/mL in RPMI 1640 medium containing 10% FBS, supplemented with 50 ng/mL of recombinant human GM-CSF and 500 U/mL of recombinant human IL4. GM-CSF (a generous gift from Schering-Plough, Dardilly, France) and IL4 were replenished after 3 days, and cells harvested at day 6 of culture. To generate IFN-DC, monocytes were cultured in the presence of 50 ng/mL GM-CSF and 1000 IU/mL IFNβ (a generous gift from Merk Serono, Rome, Italy) for 3 days and harvested at day 3 of culture (see Figure 1). 10 nM 1,25(OH)_2_D3 (a generous gift from BioXell, Milan, Italy) was added at seeding and maintained until cell collection.

### 4.3. RNA Isolation and Polymerase Chain-Reaction

Total RNA was extracted with the RNeasy Micro kit (Qiagen, Hilden, Germany) following the manufacturer’s instructions. An amount of 150–250 ng was reverse-transcribed into cDNA by M-MLV reverse transcriptase (Invitrogen) by using poly d(N)6 (GE Healthcare). Quantitative real-time PCR was performed on an ABI Prism 7500 PCR cycler (Applied Biosystems Foster City, CA 94404 USA) using SYBRgreen chemistry. Validated PCR primers for VDR (Forward 5′-ACGCCCACCATAAGACCTAC-3′, Reverse 5′-GCTGGGAGTGTGTCTGGAG-3′) and CYP24A1 (Forward 5′-ATGAGCACGTTTGGGAGGAT-3′, Reverse 5′-TGCCAGACCTTGGTGTTGAG-3′) were used. As endogenous control, primers for the housekeeping gene GAPDH (Forward 5′-GAAGGTGAAGGTCGGAGTC-3′, Reverse 5′-GAAGATGGTGATGGGATTTC-3′) were used. Relative quantification was performed by using the comparative Ct method (2^−ΔCt^), where the Ct is defined as the cycle number at which the SYBR green fluorescence crosses the threshold (arbitrary set and fixed for all experiments performed) and the Δ is the difference between the Ct of the sample tested and the housekeeping gene Ct [75,76].

### 4.4. ELISA

Soluble mediators in culture supernatants were measured by the following commercially available ELISA kits according to manufacturers’ instructions: CCL2, CCL5, sIL6R, IL8 (R&D systems, Minneapolis, MN, USA); IL-6, IL1β (Biolegend, San Diego, CA 92121, USA); TNFα, IL10 (Pierce Endogen, Rockford, IL, USA); MMP9 (ELISA search-light technology Pierce Biotechnology, Rockford, IL, USA). All biochemical measures were performed in a single batch and a comparable number of patient and control samples were always assayed simultaneously in the same ELISA plate.

### 4.5. Statistical Methods

Differences between untreated and 1,25(OH)_2_D3-treated cells were tested by two tailed Student’s *t*-test for paired samples. Donor group effects were tested by two-way ANOVA with Tukey’s post-test for multiple comparisons. Cell type effects (e.g., quantitative differences between IFN-DC and IL4-DC) were tested by the Kruskal–Wallis test with Dunn’s post-test for multiple comparisons or the Mann–Whitney test after pooling data obtained from the three donor groups. Results were considered statistically significant when *p* < 0.05. Analyses were conducted using GraphPad Prism Software version 9.4.

## Figures and Tables

**Figure 1 ijms-24-06717-f001:**
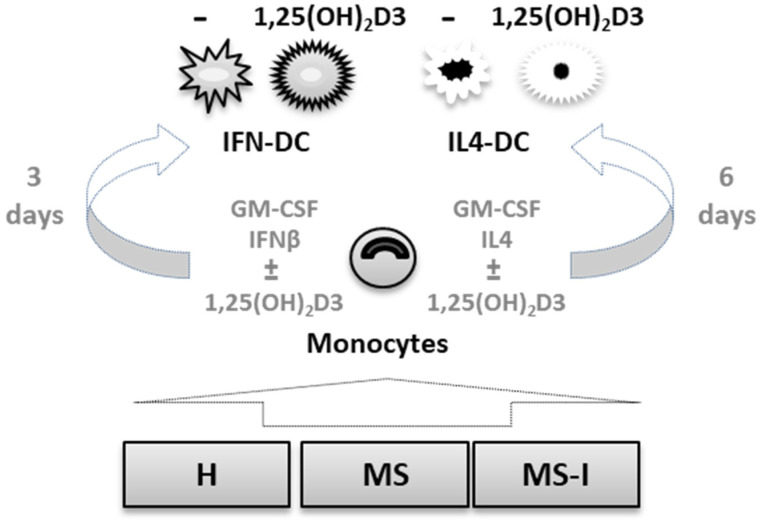
Cell models. CD14^+^ monocytes were obtained from three groups of donors: healthy donors, H; relapsing remitting multiple sclerosis (RRMS) patients free of therapy, MS; RRMS patients undergoing interferon (IFN) therapy, MS-I. Monocytes were in vitro differentiated into IFN-DC or IL4-DC, in the presence or in the absence of 10 nM 1,25-dihydroxyvitamin D3 (1,25(OH)_2_D3).

**Figure 2 ijms-24-06717-f002:**
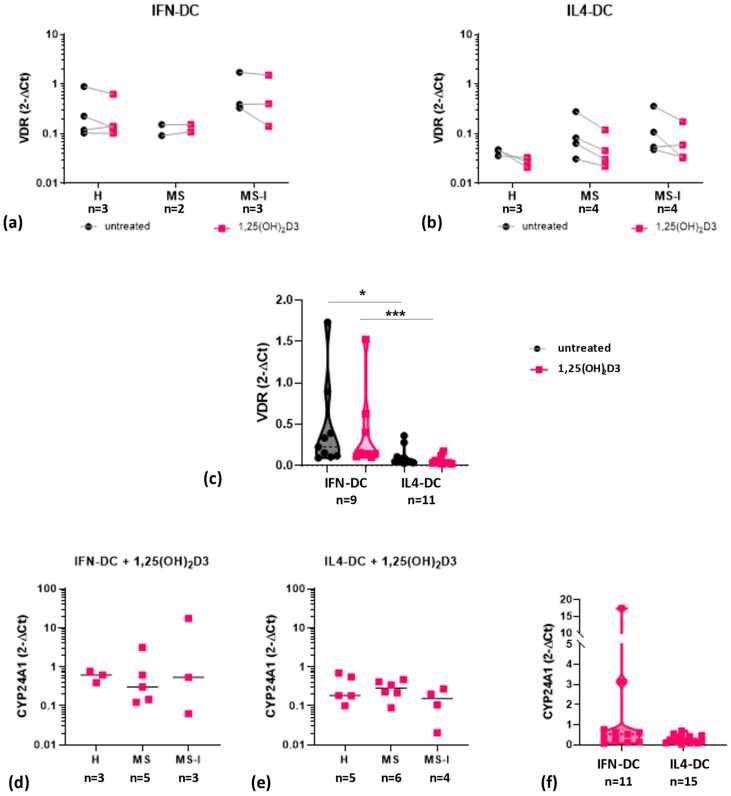
Expression and modulation by 1,25(OH)_2_D3 of vitamin D receptor (*VDR*) and *CYP24A1* mRNA. *VDR* (**a**–**c**) and *CYP24A1* (**d**–**f**) mRNA expression in IFN-DC (**a**,**d**) and IL4-DC (**b**,**e**) were analyzed by quantitative RT-PCR and expressed as 2^−ΔCT^ values. *GAPDH* was used as reference gene. Violin plot of *VDR* (**c**) and *CYP24A1* (**f**) mRNA data after pooling data obtained from the three donor groups. Cell type effects (e.g., quantitative differences between IFN-DC and IL4-DC) were tested by Kruskal–Wallis test with Dunn’s post-test for multiple comparisons for VDR and Mann–Whitney test for *CYP24A1*. Number of subjects (n) analyzed are indicated under each panel. Black dots indicate untreated cells, pink squares indicate 1,25(OH)_2_D3 treated cells; * *p* ≤ 0.05, *** *p* ≤ 0.001.

**Figure 3 ijms-24-06717-f003:**
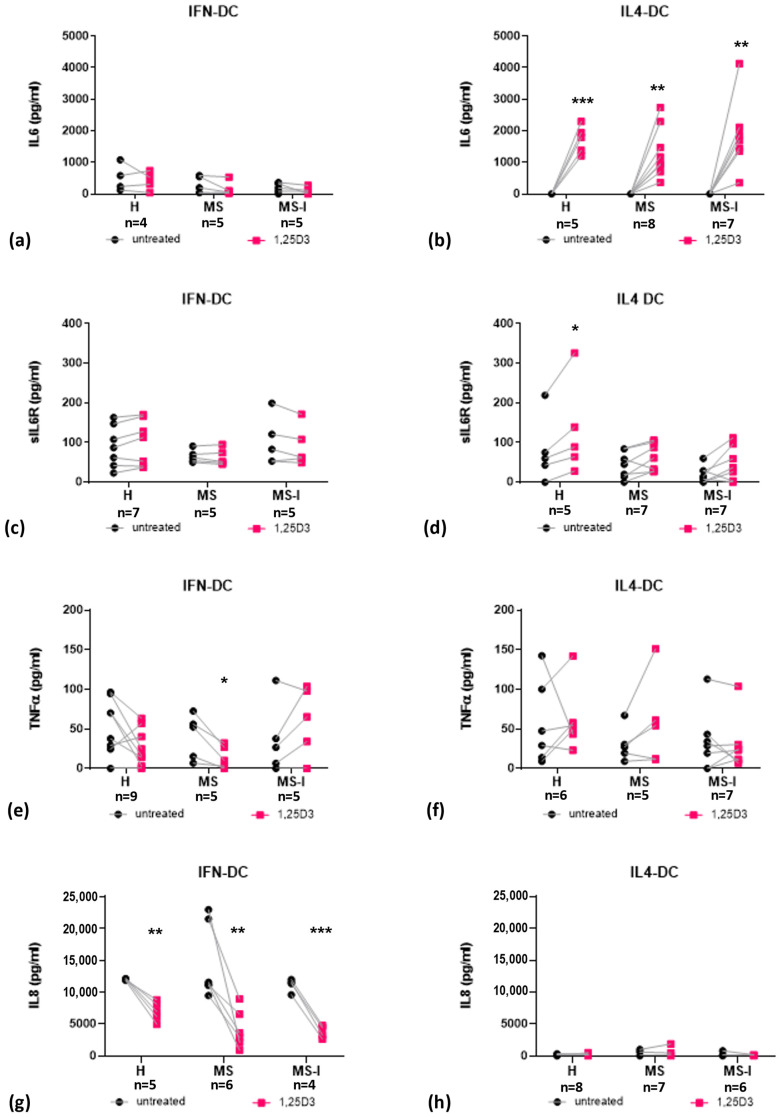
Expression and modulation by 1,25(OH)_2_D3 of pro-inflammatory cytokines. IL6 (**a**,**b**), sIL6R (**c**,**d**), TNFα (**e**,**f**) and IL8 (**g**,**h**) concentrations measured in IFN-DC (**a**,**c**,**e**,**g**) and IL4-DC (**b**,**d**,**f**,**h**) supernatants from H, MS, and MS-I donors are shown as individual values. Number of subjects (n) analyzed are indicated. Black dots represent untreated cells, pink squares represent 1,25(OH)_2_D3 treated cells. 1,25(OH)_2_D3 treatment effects were tested by Student’s *t*-test for paired samples. * *p* ≤ 0.05, ** *p* ≤0.01, *** *p* ≤ 0.001. No donor effect was found by two-way ANOVA with Tukey’s post-test.

**Figure 4 ijms-24-06717-f004:**
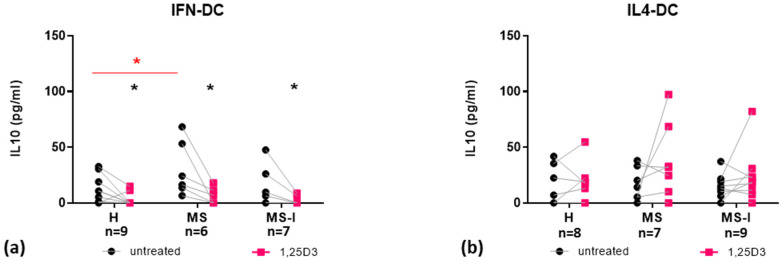
Expression and modulation by 1,25(OH)_2_D3 of IL10. IL10 concentrations measured in IFN-DC (**a**) and IL4-DC (**b**) supernatants from H, MS, and MS-I donors are shown as individual values. Number of subjects (n) analyzed are indicated. Black dots represent untreated cells, pink squares represent 1,25(OH)_2_D3 treated cells. 1,25(OH)_2_D3 treatment effects were tested by Student’s *t*-test for paired samples. Donor group effects were tested by two-way ANOVA with Tukey’s post-test for multiple comparisons. Significantly different comparison is indicated by a red bar/asterisk, * *p* ≤ 0.05.

**Figure 5 ijms-24-06717-f005:**
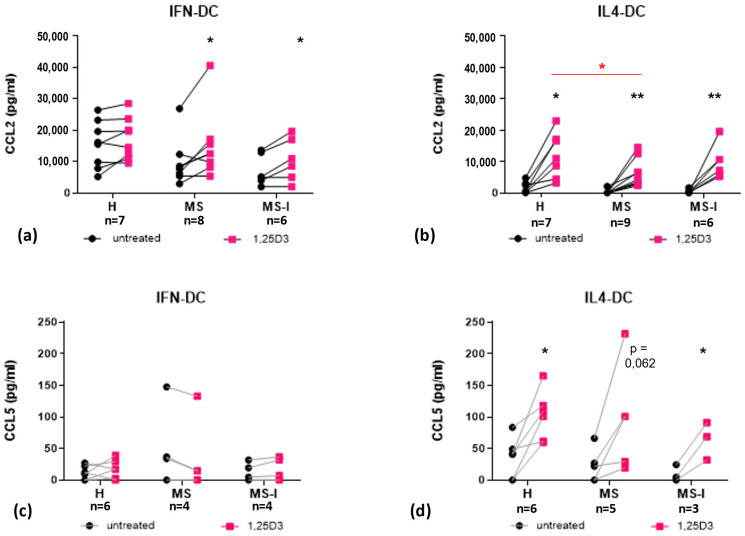
Expression and modulation by 1,25(OH)_2_D3 of CCR2 and CCR5 ligands. CCL2 (**a**,**b**) and CCL5 (**c**,**d**) concentrations measured in supernatants of IFN-DC (**a**,**c**) and IL4-DC (**b**,**d**) from H, MS, and MS-I donors are shown as individual values. Number of subjects (n) analyzed are indicated. Black dots represent untreated cells, pink squares represent 1,25(OH)_2_D3 treated cells. 1,25(OH)_2_D3 treatment effects were tested by Student’s *t*-test for paired samples (black asterisks). Donor group effects were tested by two-way ANOVA with Tukey’s post-test for multiple comparisons. Significantly different comparison is indicated by a red bar/asterisk. * *p* ≤ 0.05, ** *p* ≤ 0.01.

**Figure 6 ijms-24-06717-f006:**
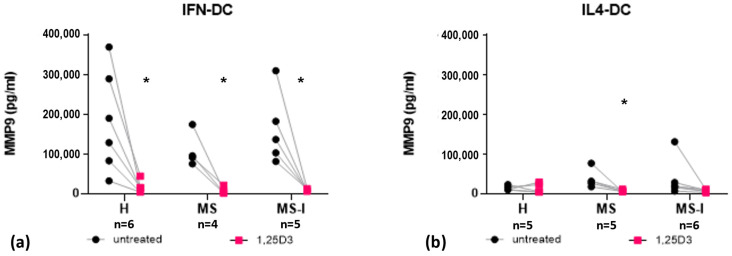
Expression and modulation by 1,25(OH)_2_D3 of matrix-metalloproteinase 9 (MMP9). MMP9 concentrations measured in IFN-DC (**a**) and IL4-DC (**b**) supernatants from H, MS, and MS-I donors are shown as individual values. Number of subjects (n) analyzed are indicated. Black dots represent untreated cells, pink squares represent 1,25(OH)_2_D3 treated cells. 1,25(OH)_2_D3 treatment effects were tested by Student’s *t*-test for paired samples, * *p* ≤ 0.05. Donor group effects were tested by two-way ANOVA with Tukey’s post-test for multiple comparisons. No significant effect was detected.

**Table 1 ijms-24-06717-t001:** Adjusted *p* values of the indicated comparisons of soluble mediators secreted in IFN-DC culture supernatants (Tukey’s post-test).

		IL6	sIL6R	TNFα	IL8	IL10	CCL2	CCL5	MMP9
**untreated**	**MS vs. H**	0.4065	0.6535	0.9357	0.3676	0.0387 *	0.3192	0.1659	0.2711
	**MS-I vs. H**	0.1846	0.9118	0.8376	0.9122	0.9967	0.1255	0.9947	0.8946
	**MS-I vs. MS**	0.8473	0.4632	0.9793	0.219	0.0614	0.7955	0.2532	0.5121
**1,25(OH)_2_D3**	**MS vs. H**	0.3271	0.3788	0.8572	0.318	0.8991	0.8471	0.5774	0.9854
	**MS-I vs. H**	0.193	0.922	0.1451	0.3011	0.9938	0.2422	0.9978	0.9884
	**MS-I vs. MS**	0.9326	0.6469	0.091	0.9808	0.8655	0.5023	0.6669	0.9996

* significant *p* value.

**Table 2 ijms-24-06717-t002:** Adjusted *p* values of the indicated comparisons of soluble mediators secreted in IL4-culture supernatants (Tukey’s post-test).

		IL6	sIL6R	TNFα	IL8	IL10	CCL2	CCL5	MMP9
**untreated**	**MS vs. H**	>0.9999	0.5604	0.4878	0.883	0.9501	0.783	0.8905	0.273
	**MS-I vs. H**	>0.9999	0.2107	0.2376	0.9975	0.9996	0.8536	0.7051	0.2509
	**MS-I vs. MS**	>0.9999	0.7309	0.9049	0.9244	0.9556	0.9971	0.9174	0.9998
**1,25(OH)_2_D3**	**MS vs. H**	0.5091	0.1504	0.9871	0.6982	0.115	0.0219 *	0.9753	0.906
	**MS-I vs. H**	0.9369	0.0607	0.1278	0.8339	0.7146	0.6219	0.4758	0.9174
	**MS-I vs. MS**	0.2502	0.8743	0.1996	0.3968	0.3817	0.2261	0.6042	0.9988

* significant *p* value.

## Data Availability

The data presented in this study are available in the article and Appendix A.

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
