# Peer review of "1,25(OH)2D3 Differently Modulates the Secretory Activity of IFN-DC and IL4-DC: A Study in Cells from Healthy Donors and MS Patients"

_ijms, 2023, doi:10.3390/ijms24076717_

Round 1

Reviewer 1 Report

In the current study, Sanseverino et al have investigated the response of IFNb/GM-CSF or IL-4 dendritic cell (DC) from 3 groups (healthy donors, RRMS patients with no therapy or with IFN-b therapy) to 1,25(OH)2D3 in vitro studies. The manuscript is well written, precise and the language is in good shape. The authors have provided relevant background introduction and references to the current study. I have only few comments that I believe would make this manuscript even better.

·      Have authors investigated how 1,25(OH)2D3 presence influences antigen specific proliferation of CD4+ T cells when cultured with IFN-DC or IL-4 DC? Or when CD4+ T cells are stimulated with anti-CD3 in the presence of IFN-DC, IL-4 DC and 1,25(OH)2D3 which might be the best suited way for checking T cell proliferation from all the three groups (healthy donors, MS patients with or without therapy).  Apart from assessing T cell proliferation, it would also give insights into expression levels of costimulatory molecules on DC.

·      Did the authors observe any difference in the survival of IFN-DC or IL-4 DC when cultured in the presence of 1,25(OH)2D3 via annexin staining.

Author Response

Response to Reviewer 1 Comments

In the current study, Sanseverino et al have investigated the response of IFNb/GM-CSF or IL-4 dendritic cell (DC) from 3 groups (healthy donors, RRMS patients with no therapy or with IFN-b therapy) to 1,25(OH)2D3 in vitro studies. The manuscript is well written, precise and the language is in good shape. The authors have provided relevant background introduction and references to the current study. I have only few comments that I believe would make this manuscript even better.

Point 1: Have authors investigated how 1,25(OH)2D3 presence influences antigen specific proliferation of CD4+ T cells when cultured with IFN-DC or IL-4 DC? Or when CD4+ T cells are stimulated with anti-CD3 in the presence of IFN-DC, IL-4 DC and 1,25(OH)2D3 which might be the best suited way for checking T cell proliferation from all the three groups (healthy donors, MS patients with or without therapy).  Apart from assessing T cell proliferation, it would also give insights into expression levels of costimulatory molecules on DC.

Response 1: We agree with the referee that such experiments would have been an important added value for the manuscript, but unfortunately the project ended up before we could set up the MLR assay. 

Point 2: Did the authors observe any difference in the survival of IFN-DC or IL-4 DC when cultured in the presence of 1,25(OH)2D3 via annexin staining.

Response 2: We did not performed annexin staining. However, we did not observe cell yield differences in the presence or absence of 1,25(OH)2D3 when cells were manually counted in tripan blue at the end of the culture.

Reviewer 2 Report

This study assessed the reactivity of two different types of DCs (IFN-DC or IL4-DC) from healthy or MS patients against 1,25(OH)2D3. It is interesting that IFN-DC and IL4-DC react with 1,25(OH)2D3 differently. For appropriate interpretation, followings need to be added in the manuscript:

(1) The frequency data of DC (CD11c positive cells) in IFN-DC and IL4-DC needs to be added. IFN-DC and IL4-DC were generated by culture for 3 and 6 days, respectively. Three days culture was enough to induce high frequency of DCs in IFN-DC?

(2) How was the concentration of 1,25(OH)2D3 set up? It was optimal for production of all tested cytokines?

(3) Title needs to be rephrased. This study did not investigate immunomodulatory function of 1,25(OH)2D3, and did examine only 1,25(OH)2D3-induced cytokine responses of DCs.

(4) Fig.2: It is good to add data about cell type effect on CYP24A1, as shown in Fig. 2c.

(5) When were the experiments performed? The use of clinical samples appeared to be approved in 2010. More than ten years ago.

(6) It is difficult to find the novelty of this study. What is a significant finding in the assessment for reactivities of DCs from healthy donor and patients with MS?

Reviewer 3 Report

The authors provide a comparison between two protocols for obtaining DCs, the classical IL-4 derived DCs and IFNb/GMCSF DCs from monocytes obtained from healthy donors, MS donors free of therapy and MS patients under IFNb therapy. They present evidence of the functional differences between DCs obtained by the two different methods and discuss their results in the context of MS pathogenesis. The methodologies applied to address the scientific question were adequate. The results are clearly presented, and the conclusion is supported by the results. Concerning the discussion, it is very extensive and there is a repetition of the results section that should be avoided. The manuscript would benefit from a more concise discussion section.

Author Response

Response to Reviewer 3 Comments

The authors provide a comparison between two protocols for obtaining DCs, the classical IL-4 derived DCs and IFNb/GMCSF DCs from monocytes obtained from healthy donors, MS donors free of therapy and MS patients under IFNb therapy. They present evidence of the functional differences between DCs obtained by the two different methods and discuss their results in the context of MS pathogenesis. The methodologies applied to address the scientific question were adequate. The results are clearly presented, and the conclusion is supported by the results. Concerning the discussion, it is very extensive and there is a repetition of the results section that should be avoided. The manuscript would benefit from a more concise discussion section.

Response 1: The discussion has been revised according to the referee’s suggestion and it is now more concise. We thank the reviewer for the suggestion, which indeed allowed to improve the readabilty of the manuscript.

Round 2

Reviewer 1 Report

Congratulations to the authors on the revised version of the manuscript. I recommend the manuscript for publication in it's current form.

Reviewer 2 Report

The authors replied to the comments by this reviewer adequately.